# Solar-Cycle Variations of Forbidden Energetic Electrons Enhancements

**Alla V. Suvorova**

Skobeltsyn Institute of Nuclear Physics, Lomonosov Moscow State University, 119991 Moscow, Russia; alla@srd.sinp.msu.ru

**Abstract:** Intense fluxes of electrons from the Earth's radiation belt (ERB) with energies of tens and hundreds of keV can penetrate to low altitudes at low latitudes outside the South Atlantic Anomaly. This region is known as a forbidden zone of quasi-trapped energetic particles. Flux enhancements of energetic electrons in the forbidden zone, so-called forbidden energetic electrons (FEE), produce significant ionization effects in the upper atmosphere at low latitudes. In this work, solar-cycle variations of the FEE enhancements with energy > 30 keV were analyzed over a 25-year period using a database of low-orbit satellites of the NOAA/POES and MetOp series. We found the highest correlations of the annual occurrence of FEE with the F10.7 solar activity index ($-0.87$) and the Alfven Mach number of the upstream solar wind (0.76). Using multiparameter regression analysis, a power expression was obtained with those parameters as well as with plasma beta and the interplanetary magnetic field strength with a total correlation coefficient of 0.94. The role of the conductivity of the high-latitude ionosphere in the mechanism of the penetration of ERB electrons into the forbidden zone is discussed.

**Keywords:** Earth's radiation belt; quasi-trapped electrons; solar cycle; Alfven Mach number; F10.7 solar activity index

## 1. Introduction

The Earth's radiation belt (ERB), a region where energetic charged particles (with energy from tens of keV to hundreds of MeV) are stably trapped by the geomagnetic field, is located above an altitude ~1000 km, extending up to 7–8 Earth's radii. However, in the vicinity of the South Atlantic Anomaly (SAA), it drops to heights of a few hundreds of km. The structure of ERB depends on the types of particles and their energy. The keV-energy electrons in the radiation belt form inner and outer zones interconnected by a slot region of very low fluxes.

The ERB particles can impact on the ionosphere causing additional ionization of atmospheric gases. Satellite studies show that energetic electrons (tens to hundreds of keV) of ERB play an important role in these processes at all latitudes. At high and middle latitudes, the impact is caused by electrons precipitating from the outer ERB during geomagnetic disturbances. Similarly, electron precipitations occur inside the area of SAA. The effects of particle precipitations in these regions have been studied in detail experimentally and theoretically for several decades [1–5]. A relatively new result in this field is the phenomenon of penetration of energetic electrons from the inner ERB to low altitudes outside the SAA area, i.e., in low-latitude regions far beyond the SAA [6,7]. This low-latitude region is known as a forbidden zone of quasi-trapped energetic particles, namely, the forbidden zone is an area of weak electron fluxes in the range of latitudes from 40° S to 40° N and longitudes from 0° to 180° E and from 100° W to 180° W (see Figure 1). The experimental evidences of the ionization effect of >30 keV electrons injected into the forbidden zone were presented in [7–11].

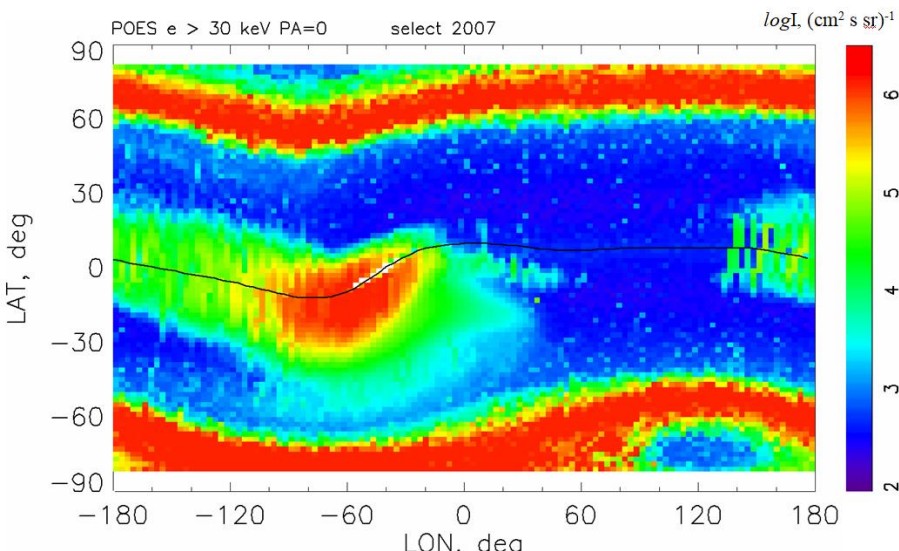

**Figure 1.** Map of maximum electron fluxes with energies >30 keV in geographic coordinates according to five NOAA/POES satellite measurements at an altitude of 850 km for one-year interval. The intensity *I* is presented in a color logarithmic scale. The magnetic equator is shown as a black curve. The projection of the outer ERB looks like bands at high latitudes (>50°) with intense electron fluxes. The projection of the inner ERB looks like a large spot in the SAA region at low latitudes in the sector of longitudes from −100° to 0°. The forbidden zone is located near the equator outside the SAA region, namely, it is an area of weak electron fluxes (colors from blue to yellow) in the range of latitudes from 40° S to 40° N and longitudes from 0° to 180° E and from 100° W to 180° W.

In the forbidden zone, the electrons are quasi-trapped because, drifting eastward across the geomagnetic field, they reach the SAA in less than 20 h, where they descend to altitudes below 100 km, which leads to their thermalization due to energy losses for ionization. The phenomenon of flux increase of quasi-trapped electrons in the forbidden zone is called forbidden energetic electrons (FEE) enhancement [12]. It was proved that the source of these electrons is the inner ERB [12]. However, the mechanism of their transport to low altitudes has not yet been fully studied. Analysis of the experimental data from NOAA/POES satellites showed that the mechanism is the rapid radial ExB transport toward the Earth, which can operate due to a strong westward electric field [7]. The statistical study of FEE enhancements found unusual regularities in their occurrence on long and short time scales such as annual, seasonal, and local time variations [13]. An explanation of these features can help in understanding how the electric field can penetrate to low latitudes at low altitudes.

Here, we present a comprehensive study of the solar-cycle variation of quasi-trapped electrons with energies >30 keV using continuous low-orbit satellite observations over a 25-year period from 1998 to 2022. The study extends and significantly enriches the statistical analyses of FEE events presented in previous works [13,14]. We use enhanced statistics by adding the data during the rising phase of the 25th solar cycle (2020–2022). The method of experimental data treatment and analysis is described in Section 2. The results of analysis are presented in Section 3 and discussed in Section 4. Section 5 includes the conclusions.

## 2. Methods

### 2.1. Data

One of the tasks of the low-orbit, polar satellites of the NOAA/POES and MetOp series is monitoring of radiation conditions under the ERB [15]. Continuous observations with identical detectors were started in 1998. The data are available at https://www.ngdc.noaa.gov/stp/satellite/poes/ (accessed on 15 August 2023). The satellites have a sun-synchronous orbit with an inclination of 98° at an altitude of about 850 km that allows measuring charged particle fluxes in a fixed range of local time. Table 1 shows the period of

satellite operation and local time (LT) ranges. For several years, the NOAA/P OES satellites moved mainly in three orbital planes, covering ranges of the terminator at 6 and 18 LT, morning–evening at 9 and 21 LT, and night–day at 2 and 14 LT. To date, the number of planes has been reduced to two: terminator and morning–evening. Three European MetOp satellites move in the morning–evening plane at 9 and 21 LT.

**Table 1.** Availability of data from NOAA/POES and MetOp satellites.

| Satellite | Interval, yy | Local Time Range, hh |
|---|---|---|
| NOAA-15 | 1998–2022 | 6–18 → 7–19 |
| NOAA-16 | 2001–2014 | 2–14 → 9–21 |
| NOAA-17 | 2002–2013 | 9–21 → 7–19 |
| NOAA-18 | 2005–2022 | 2–14 → 9–21 |
| NOAA-19 | 2009–2022 | 2–14 → 7–19 |
| MetOp-A | 2014–2022 | 9–21 |
| MetOp-B | 2006–2021 | 9–21 |
| MetOp-C | 2019–2022 | 9–21 |

The NOAA/POES and MetOp satellites are equipped with identical detectors for measuring energetic electrons and protons in a wide energy range from eV to hundreds of MeV. Pairs of detectors with an orthogonal orientation allow simultaneous measuring of the fluxes of quasi-trapped and precipitating ERB particles. In the present study, we use the integral channel >30 keV of an electron detector pointed to the zenith. This channel was chosen because the >30 keV electron fluxes are very variable, which allows the collection of large statistics of FEE events for analysis.

Figure 1 shows an example of geographic maps of >30 keV electron fluxes observed during the year 2007. The technique for constructing the maps consists in representation of the maximum electron fluxes in spatial cells of $3° \times 2°$ of geographic longitude and latitude, respectively [7]. In Figure 1, the background fluxes are shown in blue scale and enhancements in colors from green to red. In the map, several distinct spatial regions are visible. At high latitudes, where the field lines are almost vertical, the detector directed to the zenith ($0°$) observes electrons precipitating from the outer zone of ERB into the loss cone and penetrating into the atmosphere. The projection of the outer ERB looks like belts at latitudes $>50°$, located symmetrically in the northern and southern hemispheres. At low latitudes, where the magnetic field lines are almost horizontal, the detector observes quasi-trapped particles that are quickly thermalized in the upper atmosphere inside the SAA region: during the eastward azimuthal drift, the drift shells of electrons descend to the heights of dense atmosphere. The projection of the inner ERB to the height of 850 km looks like a large spot in the SAA region in the sector of longitudes from $-100°$ to $0°$.

The statistics accumulated over the year revealed a high-intensity band in the forbidden zone extending to the SAA (green−yellow−red color scale). However, this is a picture gathered over a long period of time, and the quasi-trapped electrons do not form a stable permanent structure like SAA because, in less than a day, the particles drift and sink into the dense atmosphere (see Table 1 in [14]). As one can see, the latitude distribution of the FEE enhancements is symmetrical with respect to the magnetic equator because the motion of charged particles is ordered by the dipole magnetic field.

It should be noted that the maximum flux method has an advantage over the averaging procedure. Since a flux enhancement of quasi-trapped electrons has a typical duration of several minutes and occurs rarely, the averaging of data from several satellites over a long time interval leads to smoothing of intense fluxes by very low background fluxes such that the identification of FEE enhancements becomes very difficult. Therefore, we use the geographical distributions of the maximum fluxes for visual sorting and selection of FEE events.

### 2.2. Identification and Selection of FEE Enhancement Event

The identification of FEE events is based on daily geographic maps of >30 keV electron fluxes. The daily maps are used for selection of days when strong FEE enhancements exceed the background fluxes by several orders of magnitude. As a rule, the background fluxes did not exceed ~$10^2$ (cm$^2$ s sr)$^{-1}$. Thus, an FEE event is defined as all increases above $10^4$ (cm$^2$ s sr)$^{-1}$ in fluxes of quasi-trapped electrons observed in the forbidden zone on one day [13]. An example of an FEE event is shown in Figure 2a. Note that the FEE enhancement is different from precipitating electron fluxes reported, for example, by Grigoryan et al. [16]. First, the FEE enhancement has a gradual time profile of a "dome" shape, as shown in Figure 2b, with the maximum located near the magnetic equator (see Figure 2a). In contrast, the precipitating fluxes have two maxima at latitudes >10° off the equator, as shown in Figure 2c. Secondly, the intensity of precipitating fluxes rarely exceeds $10^3$ (cm$^2$ s sr)$^{-1}$, while the vast majority of FEE events are characterized by high intensities of ~$10^4$ (cm$^2$ s sr)$^{-1}$ and above [8].

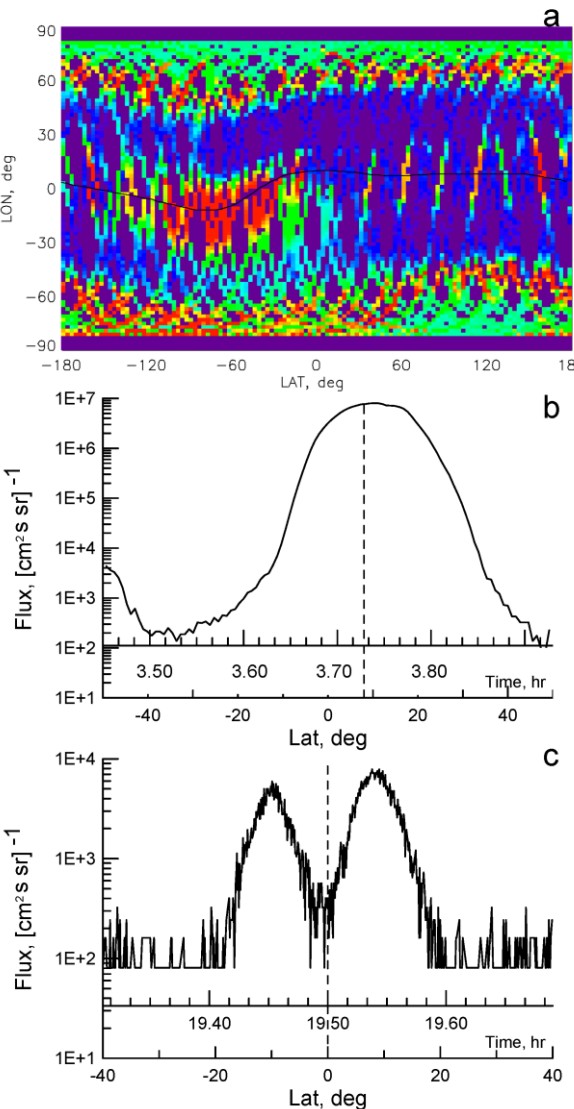

**Figure 2.** Examples of (**a**) an FEE event, which includes all enhancements during a day; (**b**) a time profile of FEE enhancement; and (**c**) a time profile of precipitating electrons. Vertical dashed line indicates magnetic equator.

The difference between the FEE enhancements (see Figure 2b) and precipitating particle events (see Figure 2c) has a deep physical meaning. In the latter case, the precipitating

particles appear at altitudes of 850 km due to strong scattering inside the inner ERB at higher altitudes and following precipitation along the field lines to the lower altitudes at low-to-middle latitudes. The precipitating particles cannot penetrate to the low-height equatorial region because they cannot move across the field lines at the magnetic equator where the magnetic field lines are horizontal. The enhancements with maximum at the magnetic equator are produced by electrons having pitch angles close to 0, i.e., the electrons originated from the stable trap region. To be observed at low altitudes, these electrons should move across the equatorial magnetic field lines from heights of the inner ERB toward the Earth without strong scattering. This earthward drift across the magnetic field lines can be produced only by an electric field in the westward direction.

As shown in Figure 2b, a single FEE enhancement has a duration of ~10 min. Different satellites can observe several enhancements at different longitudes during the day (the day is counted as one event), and sometimes this process can last several consequent days (that is, several events are counted) [9,17]. Then, the total number of such days in a year is summed up. Thus, for each year, the occurrence number of daily FEE enhancements above $10^4$ $(cm^2 \ s \ sr)^{-1}$ was determined. These numbers are used for further analysis.

## 3. Results

Statistics of FEE enhancements were compared with solar activity variations. Solar activity is characterized by the number of sunspots provided by WDC-SILSO (Royal Belgian Observatory, Brussels). The time period from 1998 to 2023 (25 years) covers the 23rd solar cycle from 1998 to 2008, the 24th solar cycle from 2009 to 2019, and the rising phase of the 25th solar cycle from 2020 to 2022. Hence, the data set acquired from the POES and MetOp satellites includes two solar maxima and two solar minima. Figure 3 shows variations in the occurrence number of FEE events per year and the yearly-smoothed monthly sunspot number over 25 years. The power of the solar cycle is determined by the sunspot number in the solar maximum. The last two cycles of the 24th and 25th are weaker in power to the 23rd. The time period under study begins in the middle of the rising phase of the 23rd cycle (1998–1999), during which few events were detected (less than 10 per year). Taking into account the fact that only one satellite was operating at that time (see Table 1), the number of events is most likely underestimated. For the rest of the interval (from 2000 to 2022), the reliability of the data is well supported statistically.

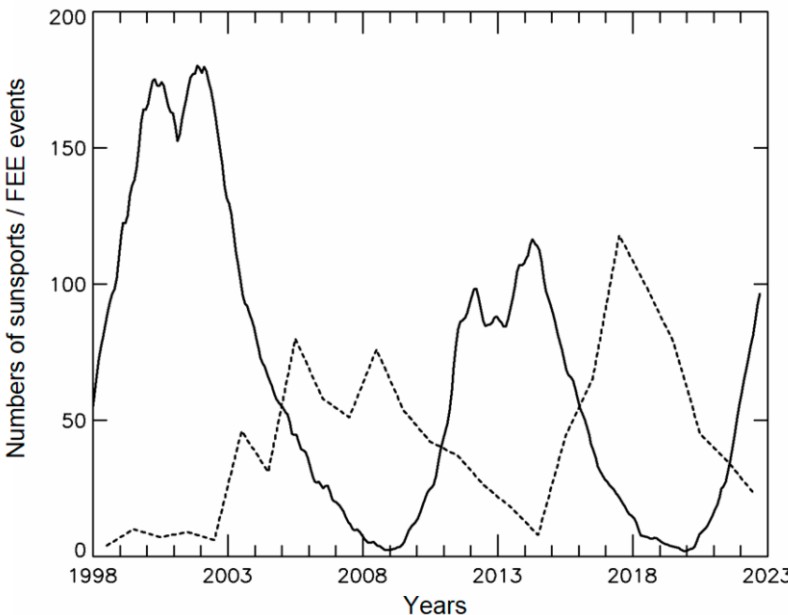

**Figure 3.** Solar-cycle variations in the annual occurrence number of FEE enhancements (dotted line) and yearly-smoothed monthly sunspot numbers (black curve).

From a comparison of the electron flux dynamics in the rising phases of the last two weak cycles, 24 and 25, we see that the annual number of events decreases from ~50 to ~30–40. In the solar maxima (2000–2001, 2013–2014), the frequency of events continues to fall, reaching a minimum number of about 10 events; the same trend is clearly visible in 2022, near the maximum of the 25th solar cycle. In the beginning of the declining phase of cycles 23 and 24 (2002 and 2014), the occurrence number of FEE events again increases rapidly over 3 years, reaching its highest values approximately after the middle of the declining phase (2005–2006 and 2017), and then begins to decrease over an interval of 8 years, including phases of minimum (2008 and 2019), rising, and maximum of solar cycles. At the same time, in the weak 24th cycle in 2017, there is an absolute maximum in the number of events over the entire period of 25 years.

It was previously shown in [14,18] that electron fluxes reach a very high intensity of more than $10^7$ (cm$^2$ s sr)$^{-1}$ during the declining phase of the 23rd and 24th cycles, where in the stronger 23rd cycle there were many more FEE events with extremely high fluxes—that is, the events in the declining phase of the 23rd cycle occurred, although less frequently, but in terms of intensity, on average, they exceeded the events in the weaker 24th cycle.

The result indicating a clear deficit in the number of electron injections during the maximum phase of solar cycles requires additional consideration. Previous results of the statistical analysis of the solar wind physical parameters during three cycles from 20 to 23 showed that two parameters, the Alfven Mach number ($M_A$) and the plasma beta $\beta$, clearly anticorrelate with the sunspot number: at the solar maximum, the number of $M_A$ and $\beta$ decrease, and at the solar minimum, they increase [19]. The data on interplanetary parameters are available at https://cdaweb.gsfc.nasa.gov (accessed on 15 August 2023). In the current work, we use these two parameters together with others such as the interplanetary magnetic field (IMF) *B* and the solar wind density *D*, velocity *V*, and temperature *T*, which vary with solar cycles in different ways [19]. Another physical parameter of solar activity is 10.7 cm radio flux (the $F_{10.7}$ index), which is commonly accepted as an indicator of the solar far ultraviolet emission affecting the ionization of the Earth's ionosphere. It is well known that $F_{10.7}$ strongly correlates with the sunspot number. Figure 4 shows solar-cycle variations of the monthly averages of the $M_A$ number, plasma $\beta$, and $F_{10.7}$ index. The average annual and monthly parameters of solar activity and solar wind are taken from the OMNI2 database [20].

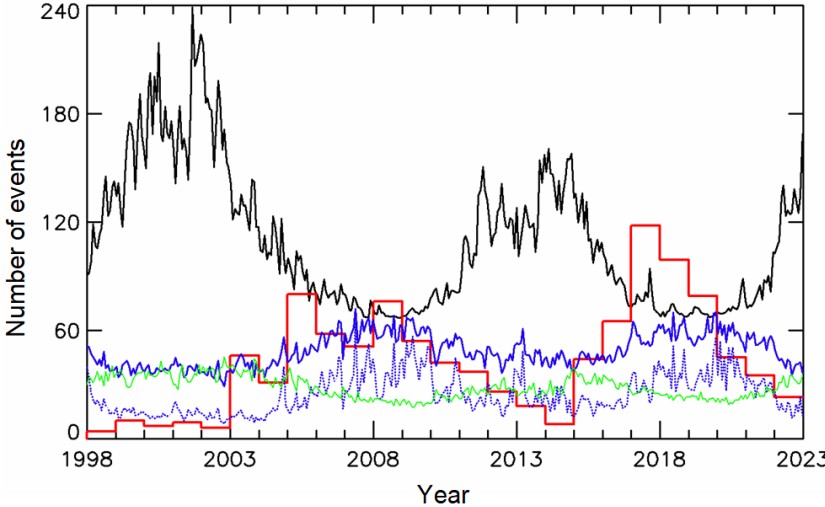

**Figure 4.** Solar-cycle variations in the annual FEE occurrence number (red histogram), the 27-day averages of the solar activity $F_{10.7}$ index (black line), and the solar wind parameters from the OMNI2 database: Alfven Mach number $M_A$*5 (blue), beta plasma $\beta$*10 (dotted blue), and IMF strength *B*\*5 (green).

We are also testing two more parameters that relate to the interaction between the solar wind and the magnetosphere. These parameters depend on the compression ratio between the densities of the solar wind and magnetosheath at the bow shock $C = D/D_{ms}$ and the local reconnection rate $R_d$ at the dayside magnetopause. They are calculated as functions of the solar wind parameters [21]. As shown in [21], the compression ratio $C$ depends on $M_A$ and is expressed as follows (Equation (7) in [21]):

$$C = \left\{ 2.44 \times 10^{-4} + [1 + 1.38\ln(M_A)]^{-6} \right\}^{-1/6} \tag{1}$$

The dayside local reconnection rate $R_d$ has a dependence on the solar wind and magnetosheath parameters, and the clock angle between the magnetic field directions in the magnetosheath and the dayside magnetosphere at the nose magnetopause. For simplicity, we have considered the case of antiparallel magnetic fields, which gives a factor equal to 1. The dayside local reconnection rate $R_d$ is expressed as follows (cf. Equation (8) in [21]):

$$R_d \propto C^{-1/2}D^{1/2}V^2\left\{1 + (M_A/6)^{1.92}\right\}^{-3/4} \tag{2}$$

In the study, a function of $M_A$ is used:

$$C* = C^{-1/2}\left\{1 + (M_A/6)^{1.92}\right\}^{-3/4} \tag{3}$$

As one can see in Figure 5, the function $C^*$ ratio has complicated nonlinear dependence on $M_A$ in the dynamic range of $M_A$ from 7 to 12, which is typical for average solar wind conditions during the 23rd and 24th solar cycles.

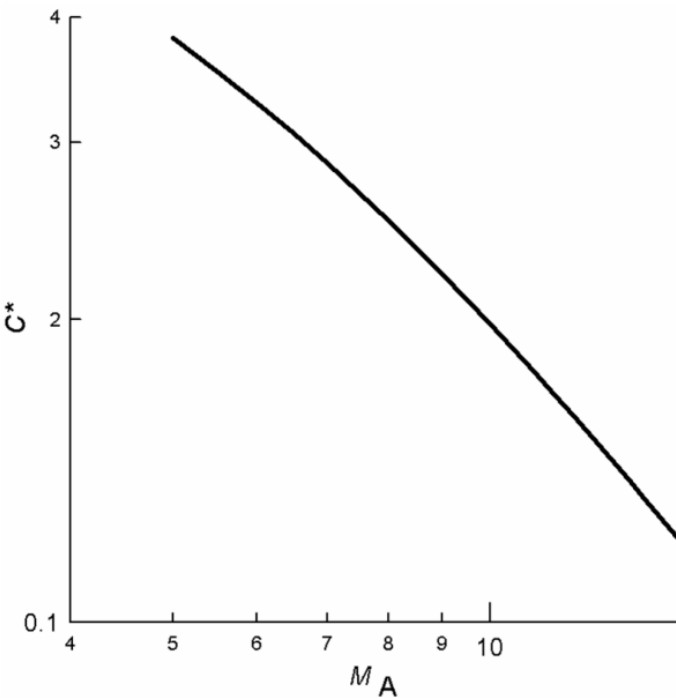

**Figure 5.** Dependence of the function $C^*$ on the Alfven Mach number $M_A$ calculated by Equation (3).

The annual averages of solar activity and solar wind parameters $M_A$, $\beta$, $D$, $V$, $T$, $B$, and $F_{10.7}$, together with the estimated driver functions $C^*$ and $R_d$, are used for correlation with the annual occurrence number of FEE events. Linear $Y = A + B \cdot X$ and power $\log Y = A + B \cdot \log X$ dependences between the FEE and the various parameters are applied in the time interval from 2000 to 2022. Figure 6 shows scatter plots with the best fits. The correlation coefficients are listed in Table 2. One can see a very poor correlation

between FEE and $D$, $V$, and $T$. However, combinations of these parameters, when used in expressions for $M_A$, $\beta$, and $C^*$, show clear power-law dependences. Linear regressions for five parameters $M_A$, $\beta$, $B$, $F_{10.7}$, and $C^*$ give good correlation coefficients. The relation between the FEE occurrence and the reconnection rate $R_d$ is weak. Multi-parametric regression analysis for combinations $(F_{10.7}, M_A, \beta, B)$, $(C^*, R_d)$, and $(B, D, V, T)$ gives total correlation coefficients 0.94, 0.88, and 0.88, respectively. Hence, the best fit is found for the following dependence:

$$\log FEE = a_0 + a_1 \log F_{10.7} + a_2 \log M_A + a_3 \log \beta + a_4 \log B \tag{4}$$

where $a_0 = 8.22$, $a_1 = -3.16$, $a_2 = 3.26$, $a_3 = -1.29$, $a_4 = 2.09$. It should be noted that the annual values of parameters in the regression (Equation (4)) are characterized by high pair correlations (see Table 2) and their variance inflation factors (VIF) are higher than 10. The high correlations are explained on the one hand by the fact that physical parameters $M_A$ and $\beta$ depend on $B$: $M_A \sim 1/B$ and $\beta \sim 1/B^2$. On the other hand, the solar parameter $F_{10.7}$ is absolutely independent from all other parameters in the physical sense. The large VIF for $F_{10.7}$ can be explained by the annual averaging such that the solar cycle variations dominate in the dynamics of solar and interplanetary parameters. The pair correlations between the parameters diminish for short-term daily or hourly variations.

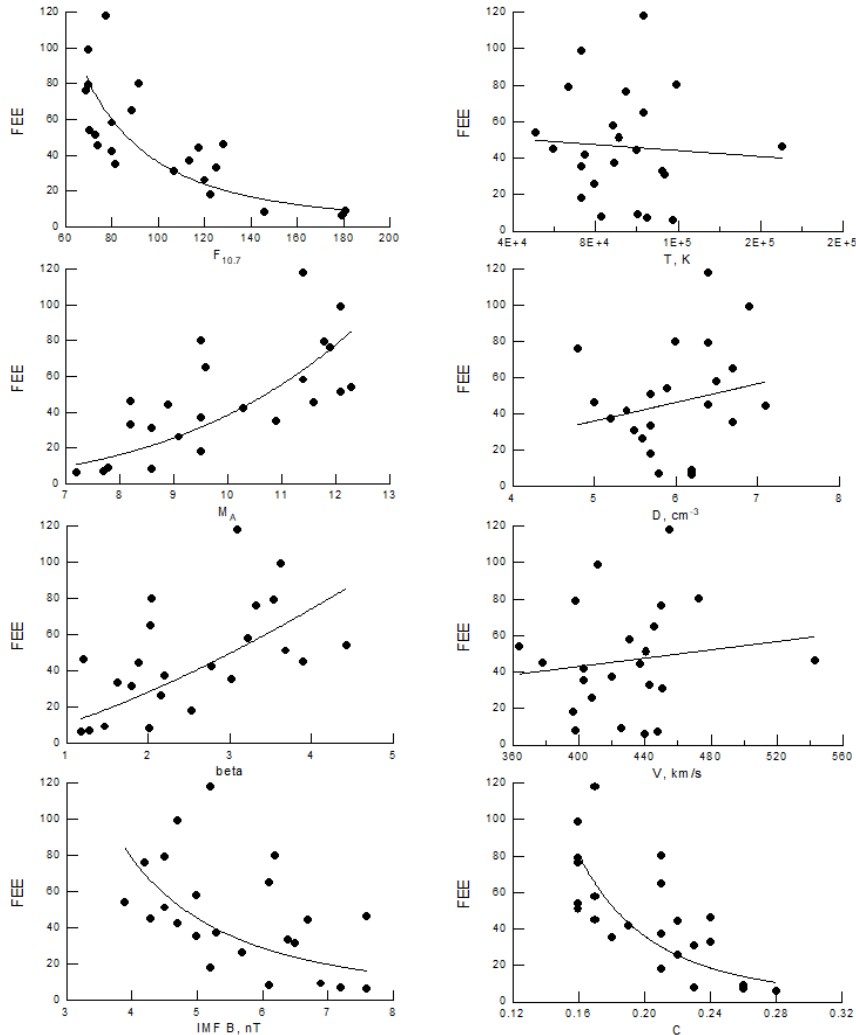

**Figure 6.** Scatter plots of the year-averages of the $F_{10.7}$ index and solar wind parameters $M_A$, $\beta$, $B$, $T$, $D$, and $V$, and of the $C^*$ function (Equation (3)) versus annual FFE occurrence number.

**Table 2.** Correlation coefficients between annual values of parameters.

|  | *FEE* | $F_{10.7}$ | *B* | *T* | *D* | *V* | $M_A$ | *β* | *C** |
|---|---|---|---|---|---|---|---|---|---|
| $F_{10.7}$ | −0.769 | | | | | | | | |
| *B* | −0.497 | 0.833 | | | | | | | |
| *T* | −0.069 | 0.434 | 0.784 | | | | | | |
| *D* | 0.209 | −0.100 | −0.020 | −0.305 | | | | | |
| *V* | 0.141 | 0.230 | 0.621 | 0.961 | −0.314 | | | | |
| $M_A$ | 0.703 | −0.905 | −0.929 | −0.610 | 0.158 | −0.402 | | | |
| *β* | 0.560 | −0.836 | −0.952 | −0.737 | 0.173 | −0.573 | 0.970 | | |
| *C** | −0.721 | 0.939 | 0.929 | 0.591 | −0.141 | 0.376 | −0.989 | −0.946 | |
| $R_d$ | −0.339 | 0.707 | 0.963 | 0.916 | −0.131 | 0.799 | −0.840 | −0.907 | 0.834 |

In the present study, the technique of multi-parametric regression is used mainly for estimation of the qualitative dependencies (correlation/anticorrelation) on the driving parameters, which have a deep physical meaning. For instance, the strong anticorrelation between *FEE* and $F_{10.7}$ can be associated with the fact that the $F_{10.7}$ index controls the ionization and, hence, conductivity of the dayside ionosphere. The physical relations of the *FEE* with the driving parameters will be discussed in the next section. Due to the small statistics, we use the criterion of chi-square ($\chi^2$) for the validation of the results presented. For the regression in Equation (4), the parameter chi-square is equal to $\chi^2 = 7.4$, which corresponds to a 99% goodness of fit for the regression of four variables and 23 data points.

We obtained that the set of physical parameters including the $F_{10.7}$ index, the $M_A$ number, the plasma *β*, and IMF *B* strongly influence the annual FEE occurrence number, while an influence of processes related to a dayside reconnection ($R_d$) during the southward IMF is weak. We emphasize an important role of the solar radio flux $F_{10.7}$, which has a high anticorrelation with the FEE occurrence.

## 4. Discussion

The effect of solar activity on the penetration of >30 keV electrons into the forbidden zone was studied in [13,14]. The accumulation of statistics allows drawing more definite conclusions for the rising phase of solar cycles. In general, 25 years of satellite measurements made it possible to reveal the following pattern in the solar-cycle variation of FEE fluxes: the occurrence number of events increases during the first 3 years of the declining phase, then decreases over the next 8 years and reaches its minimum in the solar maximum.

Previous studies reported that the intensity of FEE fluxes decreased in the 24th solar cycle, perhaps due to noticeable weakening of the interplanetary and geomagnetic sources of ERB electrons [14,18]. At the same time, the peak occurrence number increased substantially in the weak 24th cycle. This fact can be explained by an increase in the number of operating satellites from 4 to 6 (see Table 1) and by an increase in the number of single short-term (~15 min) FEE enhancements with a low intensity in the 24th solar cycle in comparison with the 23rd one—that is, injections of >30 keV electrons into the forbidden zone, although not so intense, continued even when solar wind drivers and, accordingly, geomagnetic activity weakened. The latter circumstance possibly indicates a change in the internal conditions associated with the Earth's magnetic field at low latitudes, as discussed in detail in [22,23].

An analysis of the distribution of quasi-trapped electrons at low latitudes at an altitude of 850 km [13] gave reliable confirmation of the mechanism of radial transport from the inner ERB. We believe that the main agent in this mechanism is the induced westward electric field in the post-midnight sector because it is in the sector of 2–6 LT that the highest probability of electron injections was found [13]. According to a rough estimate [7], this requires an electric field strength of at least 2 mV/m. Other researchers simulated the observed injection during a strong magnetic storm using a simplified model of the effect of an enhanced convection electric field on the inner ERB and obtained a field value of 5 mV/m [24–26]. Note that the model calculations were limited by altitudes above 2000 km,

i.e., within the inner ERB region. However, which electrodynamic process leads to the appearance of sufficiently strong (several mV/m) convection electric field at heights less than 2000 km is still an open question. Additionally, note an interesting fact that many FEE events observed in the absence of geomagnetic storms often occurred during the solar minimum [27]. The authors found that the trigger for such events is local pressure pulses associated with the dynamics of the foreshock in front of the Earth's bow shock.

On the other hand, the study by Ilie et al. [28] presented a theoretical approach to calculate the inductive electric field, which is generated by a localized change in the magnetic field. It is important to note that the effect of the inductive electric field is global—that is, it extends over all space, even if changes in magnetic field are localized and occur at a distance of several Earth radii. Hence, in the near-Earth region, the radial transport of trapped particles into the forbidden zone can be produced by a strong azimuthal inductive electric field. A huge variety of disturbing parameters of solar wind can explain FEE enhancements occurring during both geomagnetic activity and quiet time. Disturbances in the geomagnetic filed can be generated by numerous different variations of solar wind parameters. As for experiments to measure the electric fields in the inner magnetosphere, Burke and Maynard [29] showed that measurable strong electric fields can penetrate in the equatorial plane at least to the distance of two Earth's radii; however, they noted that the penetration may be extended beyond the range of reliable measurements by the instruments. Thus, there is still no evidence of strong electric fields at altitudes below 2000 km due to limitations of experimental techniques.

The present study shows a clear anticorrelation of FEE occurrence with the solar cycle ($W$ & $F_{10.7}$ indices). This dynamic is similar to the solar cycle variation of galactic cosmic rays (GCR) as reported in [30]. Previous studies showed that the interaction of GCR with the atmosphere produces high-energy (~200–800 keV) electrons via a β-decay of albedo neutrons [31]. Those electrons can be trapped near the lower edge of the inner ERB. Hence, GCR might be considered as a potential source of FEE. However, the fluxes of albedo neutrons and secondary electrons at equatorial latitudes are much lower (orders of magnitudes) than the fluxes of FEE. Moreover, the latitudinal profile of particles generated in the decay should be positive, i.e., the flux of secondary particles increases with latitude due to a decrease of the cut-off rigidity of incident cosmic rays. Figure 2 demonstrates totally different pattern for the FEE latitudinal profiles. Finally, the FEE events show clear seasonal and local time variations in the frequency of occurrence [13] which is not observed in the GRC modulation. Hence, the FEE events cannot be related to the effect of GCR.

In order to explain the minimum occurrence number of FEE events in the solar maximum, we analyzed solar wind parameters, which anticorrelate with the solar cycle. We found two parameters $M_A$ and $β$, which demonstrate high correlation with the annual numbers of FEE events. The $M_A$ number is known to be one of the key parameters controlling the effect of solar wind on the magnetosphere; so, geomagnetic activity at high latitudes anticorrelates with $M_A$ [32]. We believe that this fact can be an important argument in favor of the mechanism of electric field penetration from high latitudes to the equator. This can be understood in the following way. Substorms, as a manifestation of geomagnetic activity at high latitudes, result in enhanced precipitation of auroral particles at night. Note the substorms are characterized by intense precipitations of large scale and long duration, which change dramatically the ionization and conductivity in the high-latitude ionosphere on the night side. High-latitude precipitations also can occur on the day side, in the vicinity of the cusp for example [33], but they are local in nature with a much lower intensity and short duration that result in a weak influence to the auroral ionosphere.

During the maximum of solar activity, when the $M_A$ number is low and substorm recurrence period is short [32], the ionization of nighttime ionosphere increases on average. Hence, the enhanced conductivity in the nightside auroral region restrains the penetration of the electric field from high latitudes to the equator. In the opposite case, when $M_A$ is large (solar minimum) or increases (rising and declining phases), the substorm recurrence period is longer [32]; hence, this condition (the smallest ionization at nighttime, especially in early

morning) promotes the penetration of the electric field into low latitudes on the nightside. Thus, the mechanism of the radial transport of the inner ERB electrons into the forbidden zone due to the penetrated electric field effectively operates under the high-$M_A$-number solar wind and does not under the low-$M_A$-number. We emphasize the important role of the nighttime ionospheric conductivity in this mechanism.

Previously, the important role of the high-latitude ionospheric conductivity was also found in the seasonal variations of the FEE occurrence rate [13]. In Figure 7, one can clearly see a maximal FEE occurrence rate during northern summer periods, from May to September, and a secondary maximum corresponding to southern summer months, from December to February. Study [13] clearly demonstrated that the seasonal feature in the FEE occurrence rate resulted from the changes of the auroral ionosphere conductivity due to a combination effect of the dipole tilt angle with solar illumination and to the fast solar wind velocity. Note that the higher the solar wind velocity, the greater the $M_A$ number (for fixed values of $D$ and IMF $B$)—that is, in seasonal variations, also a larger value of $M_A$ contributes to the occurrence of FEE events.

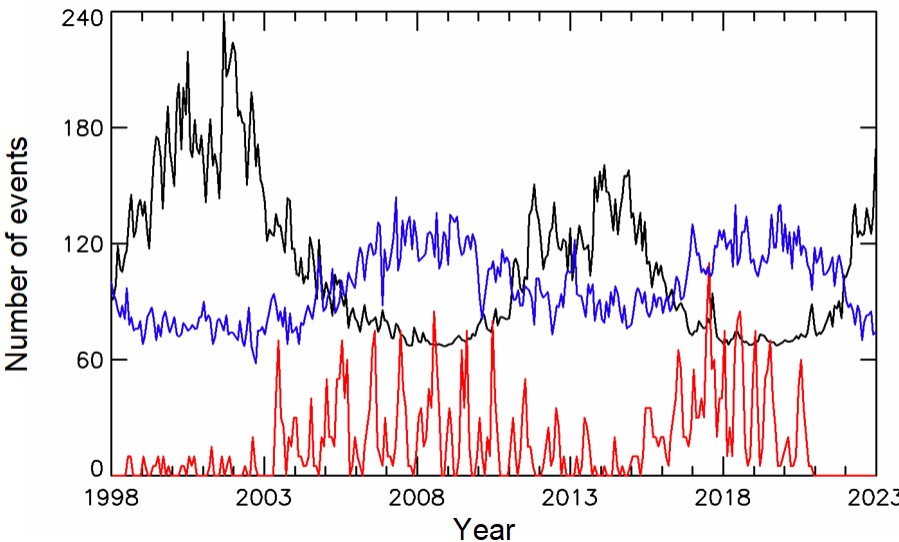

**Figure 7.** Solar-cycle variations in the monthly FEE occurrence number (red), the 27-day averages of the $F_{10.7}$ index (black line), and the Alfven Mach number $M_A$*5 (blue).

Another physical parameter of the solar activity we tested is the 10.7 cm radio flux (the $F_{10.7}$ index). As is known, the $F_{10.7}$ index correlates with the solar cycle and influences the dayside ionospheric ionization [34,35]. We found that FEE occurrence strongly anticorrelates with $F_{10.7}$, which is the high occurrence of the annual number of FEE events associated with low annual values of the $F_{10.7}$ index. In terms of ionospheric conductivity, it means that during solar minimum the ionization of the ionosphere is significantly lower compared with that in solar maximum. The $F_{10.7}$ index controls the ionization and, hence, conductivity of the dayside ionosphere such that the ionization in the northern hemisphere has a maximum in May and June [35]. The ionospheric ionization and conductivity decrease dramatically in the nighttime, especially in the early morning, and they have a poor correlation with the $F_{10.7}$ index [34]. Apparently, the nightside ionization of the high-latitude ionosphere is controlled by auroral activity, which anticorrelates with $M_A$.

Thus, a combination of the following factors might create the condition triggering the FEE enhancements: (1) strong illumination and high conductivity of the auroral region, especially during summer months; (2) very low conductivity in the sub-auroral region at night and early morning local time. A greater difference between the day and night side conductivities promotes more effective penetration of perturbations in the electric field at high latitudes into the inner magnetosphere [13].

## 5. Conclusions

The statistical analysis was conducted for fluxes of energetic electrons measured by the sun-synchronous, low-orbit, polar satellites of the NOAA/POES and MetOp series at a height of ~850 km over a long period of 25 years. The occurrence of flux enhancements of the >30 keV electrons in the forbidden zone (FEE enhancements) shows a clear solar-cycle variation. From 25 years of data, the following pattern in the solar-cycle variation of FEE enhancements was revealed: the annual occurrence number increases during the first 3 years of the declining phase and decreases over the next 8 years. Thus, the highest frequency of FEE events is observed during the second half of declining phases of the solar cycle, while the occurrence number reaches its minimum during the solar maximum.

We have accomplished the correlation and multiparameter regression analysis of the annual FEE occurrence number versus various solar activity and solar wind parameters. High correlations were found with the F10.7 solar index ($-0.87$) and the Alfven Mach number of the upstream solar wind (0.76). The best fit was obtained in the form of power dependence on $F_{10.7}$, $M_A$, $\beta$, and $B$ with the total correlation coefficient of 0.94.

These results allow us to conclude that the conductivity of the mid- and high-latitude ionosphere at nighttime plays a very important role in the mechanism of rapid radial transport of the ERB electrons into the forbidden zone. The low-conductance nighttime ionosphere is a critical condition for the high-latitude electric field to penetrate to low latitudes in the local time sector of 2–6 h. The penetration of a strong electric field of several mV/m at heights below 2000 km can provide rapid transport of the ERB energetic electrons to the forbidden zone at heights ~800 km.

**Funding:** This research received no external funding.

**Data Availability Statement:** All the data used in this study are publicly available. The NOAA/POES data were obtained from https://www.ngdc.noaa.gov/stp/satellite/poes/, accessed on 15 May 2023. The solar activity indices were obtained from NASA Goddard Space Flight Center's CDAWeb https://cdaweb.gsfc.nasa.gov, accessed on 15 May 2023.

**Acknowledgments:** The author is grateful to the NOAA team for providing experimental data on energetic particles measured by the POES и MetOp satellites. The author is grateful to the CDAWeb team for providing the free public database OMNI2 of the solar wind.

**Conflicts of Interest:** The author declares that there is no conflict of interests regarding the publication of this paper.

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
