# Peer review of "Solar-Cycle Variations of Forbidden Energetic Electrons Enhancements"

_universe, doi:10.3390/universe9080374_

Round 1
Reviewer 1 Report
Reviewer's report ID universe-2474821
Title: Solar-cycle variations of forbidden energetic electrons enhancements
Date: June 16, 2023
The manuscript is linked to evaluate the association between the annual forbidden energetic electrons and solar wind parameters. The solar activity index (F10.7), the Alfven Mach number, the plasma beta, the interplanetary magnetic field strength, the solar wind density, velocity, and temperature were used as factors. There are some questions about the structure of the manuscript and statistical analysis.
The comments about the manuscript:
1. In the methods chapter, it must be specified what data was used for the analysis and where it was taken from (indicate internet addresses). The methods section will describe the statistical methods used in the manuscript.
2. The statistical significance of correlation (Table 2) and regression coefficients (formulae (4)) should be presented. In the multivariate linear regression, the multicollinearity between the factors should be tested (I would suggest using the VIF factor). Since space weather variables used in the analysis are highly correlated, I would suggest presenting a correlation matrix between annual space weather variables.
Conclusion: A major revision is necessary.
Reviewer 2 Report
The paper is generally well written and nearly publishable. Below are
my specific comments.
1) page 1, line 37-38, "Relatively new result ... penetration of
energetic electrons ... to low altitudes /outside/ the SAA area." What
does "outside" mean in this context? Does it mean "everywhere except
in SAA" or "in a ring-shaped region at the perimeter of the SAA"? In
the Asikainen-Mursula (2005) paper they talk about stormtime
enhancement of electron flux /in/ the SAA (and also somewhat outside of
it). The same word "outside" appears also in the Abstract.
2) page 1, line 39-40, "This low-latitude region is known as the
forbidden zone..." Please define the forbidden zone precisely. Is it
everything in the latitude range ~ -60 .. +60 in Figure 1? Or is it
only the blue region in Figure 1?
3) page 2, line 49, "... due to strong electric field". Of which
direction, in order to cause radial ExB drift? Radial inward or
outward or both?
4) page 7, line 218, "... very pure correlation ...", should be "poor"?
5) Figure 6, please list the parameters in the caption in the same
order that they appear in the plot itself. If you can, please add the
subscripts in the x-axes of the plots for D, T, V, etc. Probably this
is doable, because you have already done it for F10.7.
6) I was left thinking if the good fit of equation (4) is a deep
result or if it's just that by adding enough parameters in the
correlation, sooner or later one finds expression with a high
correlation coefficient. In other words, what is the statistical
significance of the fit. In the Conclusions and Abstract you quote the
correlation coefficient result (0.94), but without critical analysis
of its significance. The paper would become stronger if you could also
assess somehow quantitatively the statistical significance (or
insignificance) of this result.
Author Response
We thank the Reviewer #2. The comments and suggestions were very useful for improvements the manuscript.
Comment 1) page 1, line 37-38, "Relatively new result ... penetration of energetic electrons ... to low altitudes /outside/ the SAA area." What does "outside" mean in this context? Does it mean "everywhere except in SAA" or "in a ring-shaped region at the perimeter of the SAA"? In the Asikainen-Mursula (2005) paper they talk about stormtime enhancement of electron flux /in/ the SAA (and also somewhat outside of it). The same word "outside" appears also in the Abstract.
Reply
Yes, this statement means “everywhere except in SAA”. Though, the fluxes inside SAA also increase during some magnetic storms. The term “in a ring-shaped region at the perimeter of the SAA” is close to “in vicinity of SAA” rather than “outside SAA”.
The sentence has been revised accordingly:
“Relatively new result in this field is the phenomenon of penetration of energetic electrons from the inner ERB to low altitudes outside the SAA area, i.e., in low-latitude regions far beyond the SAA [6, 7].”
Comment 2) page 1, line 39-40, "This low-latitude region is known as the forbidden zone..." Please define the forbidden zone precisely. Is it everything in the latitude range ~ -60 .. +60 in Figure 1? Or is it only the blue region in Figure 1?
Reply
The following sentence has been added for the definition:
“Namely, the forbidden zone is an area of weak electron fluxes in the range of latitudes from 40°S to 40°N and longitudes from 0° to 180°E and from 100°W to 180°W (see Figure 1).”
Comment 3) page 2, line 49, "... due to strong electric field". Of which direction, in order to cause radial ExB drift? Radial inward or outward or both?
Reply:
Revised: “rapid radial ExB transport toward the Earth which can operate due to a strong westward electric field”
Comment 4) page 7, line 218, "... very pure correlation ...", should be "poor"?
Corrected
Comment 5) Figure 6, please list the parameters in the caption in the same order that they appear in the plot itself. If you can, please add the subscripts in the x-axes of the plots for D, T, V, etc. Probably this is doable, because you have already done it for F10.7.
Reply
The subscripts “SW” for the key solar wind parameters have been removed in the whole text.
The caption of Figure 6 has been revised: “Figure 6. Scatter plots of the year-averages of the F10.7 index and solar wind parameters MA, β, B, T, D, V, and the C* function (eq.3) versus annual FFE occurrence number.”
Comment 6) I was left thinking if the good fit of equation (4) is a deep result or if it's just that by adding enough parameters in the correlation, sooner or later one finds expression with a high correlation coefficient. In other words, what is the statistical significance of the fit. In the Conclusions and Abstract you quote the correlation coefficient result (0.94), but without critical analysis of its significance. The paper would become stronger if you could also assess somehow quantitatively the statistical significance (or insignificance) of this result.
Reply
A criterion of chi-square is used for quantitative validation. The following sentence is added after the Eq. 4: “The parameter chi-square is equal to c2 = 7.4 that corresponds to 99% goodness of fit for the regression of 4 variables and 23 data points.”

Reviewer 3 Report
The paper is well written and clear enough. The introduction is well described, as is the analysis carried out. I have just a few comments:
Line 80 – Could the author explain the reason of the >30keV energy channel?
Line 101 – An indication of the forbidden zone in the Figure would be appreciated
Figure 2 – The X-axis with the time scale is not clear.
Line 127-128 – The author should comment more on the difference between FEE enhancement events and precipitating particle events. In particular the different shapes visible in the temporal profile should be explained
Author Response
We appreciate the Reviewer #3 for very useful comments. The manuscript has been revised accordingly.
Comment
Line 80 - Could the author explain the reason of the >30keV energy channel?
Reply
The following sentence is added here for explanation:
“This channel was chosen because the > 30 keV electron fluxes are very variable that allows collecting large statistics of FEE events for analysis.”
Comment
Line 101 - An indication of the forbidden zone in the Figure would be appreciated
Reply
The sentence has been revised accordingly:
“The forbidden zone is located near the equator outside the SAA region. Namely, it is an area of weak electron fluxes (colors from blue to yellow) in the range of latitudes from 40°S to 40°N and longitudes from 0° to 180°E and from 100°W to 180°W.”
Comment
Figure 2 - The X-axis with the time scale is not clear.
Reply
Thank you for the notice. The time in Figure 2b has been corrected.
Comment
Line 127-128 - The author should comment more on the difference between FEE enhancement events and precipitating particle events. In particular the different shapes visible in the temporal profile should be explained
Reply
The following paragraph has been added in order to explain this important issue:
“The difference between the FEE enhancements (see Figure 2b) and precipitating particle events (see Figure 2c) has a deep physical meaning. In the latter case, the precipitating particles appear at altitudes of 850 km due to strong scattering inside the inner ERB at higher altitudes and following precipitation along the field lines to the lower altitudes at low to middle latitudes. The precipitating particles cannot penetrate to the low-height equatorial region because they cannot move across the field lines at the magnetic equator where the magnetic field lines are horizontal. The enhancements with maximum at magnetic equator are produced by electrons having pitch angles close to 0, i.e., the electrons originated from the stable trap region. To being observed at low altitudes, these electrons should move across the equatorial magnetic field lines from heights of the inner ERB toward the Earth without strong scattering. This earthward drift across the magnetic field lines can be produced only by an electric field of westward direction.”

Round 2
Reviewer 1 Report
According to the recommendations of statisticians (for example https://www.investopedia.com/terms/v/variance-inflation-factor.asp) , any variable with a high VIF value (above 5 or 10) should be removed from the model. This leads to a simpler model without compromising the model accuracy, which is good.
Note that, in a large data set presenting multiple correlated predictor variables, you can perform principal component regression and partial least square regression strategies.
However, values of the VIF of 10, 20, 40, or even higher do not, by themselves, discount the results of regression analyses, call for the elimination of one or more independent variables from the analysis, suggest the use of ridge regression, or require combining of independent variable into a single index. (O'Brien, R. M. (2007). A caution regarding rules of thumb for variance inflation factors. Quality & Quantity: International Journal of Methodology, 41(5), 673–690. https://doi.org/10.1007/s11135-006-9018-6).
In the text, authors should be explaining why they presented the regression model with high values of VIP (this was done in the reply to the reviewer).
Author Response
The reply is attached
